# Immediate and short-term effects of single-task and motor-cognitive dual-task on executive function

**Weibin Zhang[1], Hua Liu[2]\*, Tong Zhang[3]**

**1** School of Sports Medicine and Rehabilitation, North Sichuan Medical College, Gaoping District, Nanchong, Sichuan Province, China, **2** School of Kinesiology and Health, Capital University of Physical Education and Sports, Haidian District, Beijing, China, **3** Rehabilitation Teaching and Research Department, Zhoukou Vocational and Technical College, Zhoukou City, Henan Province, China

\* liuhua@cupes.edu.cn

**Data Availability Statement:** The data that support the findings of this study are available from figshare, DOI: 10.6084/m9.figshare.23177645.

**Funding:** This work is a part of the project of "the relationship between physical activity and executive

## Abstract

### Objectives

Executive function plays an important role in our daily life and can be affected by both single task (acute aerobic exercise or cognitive training) and dual-task (acute motor-cognitive training) interventions. Here we explored the immediate and short-term effect on executive function to texted whether dual-task interventions are more effective at promoting executive function.

### Methods

Forty-six young men were recruited (mean age: 20.65 years) and assigned randomly to aerobic exercise (n = 15), cognitive training (n = 15), or dual-task (n = 16) groups. Executive functions were assessed before, immediately after, and 30 min after intervention using Go/No-go, 2-back, and More-Odd-Shifting tests.

### Results

Working memory function improved after all three interventions (significant Time effect, $F_{(2,86)} = 7.05$, $p = 0.001$). Performance on the 2-back test was significantly better immediately after dual-task intervention ($p = 0.038$) and the response time was shorter ($p = 0.023$). Performance on the More-Odd-Shifting test improved over time (significant Time effect, $F_{(2,86)} = 30.698$, $p = 0.01$), both immediately after the dual-task intervention ($p = 0.015$), and 30 min later ($p = 0.001$). Shifting-test performance was also better immediately after ($p = 0.005$) and 30 min after ($p < 0.001$) aerobic exercise.

### Conclusion

Executive function was enhanced by single-task (acute aerobic exercise or cognitive training) and dual-task interventions. The effect continued for 30 min after both the single-task aerobic exercise and the dual-task intervention. For short-term intervention, the dual-task was not more effective than either of the single tasks.

function and academic performance in colleges students" by North Sichuan Medical College, China (CBY21-QB14). The funders had no role in study design, data collection and analysis, decision to publish, or preparation of manuscript.

**Competing interests:** the author/s claimed no conflict of interest in the present study.

## 1. Introduction

In daily life, people must often perform motor and cognitive tasks at same time, such as shopping, counting coins while queuing up, or walking while answering a mobile phone call. Poor dual-task performance can greatly impact activities of daily life [1]. Performance in dual-task situations requires executive attention and simultaneous control of motor and cognitive functions and tends to decline with age [2]. Executive function (EF) refers to high-level cognitive ability that is used for controlling and regulating lower-level cognitive processes. EF consists of three types of abilities: inhibition control (inhibition), information updating (updating), and mental set shifting (shifting) [3]. Inhibition control is the ability to voluntarily suppress task-irrelevant responses in favor of task-relevant goal-directed responses [4]. Mental set shifting is the ability to switch attention between multiple tasks, operations, or mental sets [5]. Information updating involves continuously modifying the content of working memory as new information comes in [6].

EF continues to change throughout life [7] and the decline appears earlier than people think [8]. In modern society, sedentary life styles [9], depression [10], obesity, and being overweight [11] may lead to impaired EF. Therefore, exploring ways to improve EF is necessary. The most common strategies for increasing individual EF are non-pharmacological [12].

Aerobic exercise is a good non-pharmacological means of improving EF [13]. Takahashi et al. [14] found that only 10 min of badminton can improve EF in young people. Further study found that EF improved during the exercise-recovery period (10 min later), but not immediately after exercise [15]. However, although aerobic exercise seems to be an easy and economical therapeutic intervention, the effects vary due to gender differences [16], exercise protocols, and age [17]. Another non-pharmacological means of improving EF is through cognitive training. Studies indicate that long-term cognitive training improves EF [18, 19], however, some issues remain unclear. For example, in one study, cognitive training failed to improve inhibition control [20]. However, the participants in this study were older individuals who already have lower levels of cognitive function. This is an important confounding factor when analyzing the effects of cognitive training [21]. Additionally, a rare study showed the immediate effect of cognitive training on EF.

Motor-cognitive dual-tasks might be more effective than single-domain tasks as a way to improve executive function because dual tasks have been reported to use greater cognitive resources than the same tasks done separately (aerobic exercise or cognitive training alone) [22]. However, the findings are mixed, with many factors leading to the inconsistent results. In one study, improvements in EF after dual-task training were higher in older people than in young people [23]. Thus, the purpose of the current study was to determine how improvements in EF differ between dual-task and single-task interventions, and to ascertain the immediate and short-term effects. This knowledge will allow us to build a framework for future research into high-efficiency approaches for increasing EF in people with a variety of backgrounds. To this end, we recruited young males who were not cognitively impaired and placed them into one of three intervention groups.

## 2. Methods

### 2.1 Participants

We estimated simple size (n) using a power analysis with the following parameters in G*power. We set effect size = 0.25; alpha err prob = 0.05; power (1—beta err prob) = 0.80; number of groups = 3; number of measurements = 3; corr among rep measures = 0.5; and nonsphericity correction $\varepsilon$ = 0.5. The resulting sample size (n) was 45. This work was

conducted using a small sample size because recruiting participants was difficult during the COVIN-19 period.

Participants were generally healthy young men who were recruited through social media and telephone advertisements. Inclusion criteria were as follows: (1) Community-dwelling young men aged 18–30 years; (2) no cognitive impairments; (3) Montreal Cognitive Assessment (MoCA) score ≥ 24; (4) ability to read Chinese characters; (5) normal or corrected-to-normal vision; (6) had enough sleep the day before the experiment; (7) middle to high physical activity levels as determined by the International Physical Activity Questionnaire (IPAQ); (8) received a good education (high school degree and above), could understand simple English words. Exclusion criteria were as follows: (1) color blindness or deficiency in color vision; (2) any severe cardiopulmonary disease; (3) any musculoskeletal difficulties that prevent exercise or other movement disorders; (4) taking of any medication during the intervention period that could impair cognitive function; and (5) alcohol consumption within 24 hours before the experiment. Individuals who met the inclusion criteria provided their informed written consent to participate in this study.

## 2.2 Interventions

Participants were assigned to one of three groups: aerobic exercise, cognitive training, or dual-task (both aerobic exercise and cognitive training). Interventions lasted 30 min and were conducted in a quiet room. **Table 1** (Table 1 The intervention procedures) presents a summary of the intervention program.

**2.2.1 Aerobic exercise intervention.**   Based on the American College of Sports Medicine's standard for grading the intensity of aerobic exercise for healthy adults, combined with the research results from other scholars, participant heart rate while exercising should range between 60% to 70% of the maximum heart rate [24, 25]. The aerobic exercise intervention was performed on a stationary bicycle ergometer (ergoline, 100K, Germany). First, participants warmed up by pedaling for 5 min. The primary intervention began when the heart rate reached 60% to 70% of the maximum heart rate (calculated by subtracting participant age from 220) and lasted 20 minutes [26]. The heart rate was continuously recorded using a chest strap heart-rate monitor (Polar FT1, Polar Electro Oy, Finland). Participants took a 5-min rest after the primary intervention.

**2.2.2 Cognitive intervention.**   Based on earlier studies [27], four cognitive training tasks were included to improve executive function: the naming task, a calculation task, the Stroop color-word task, and a working memory task.

**Table 1.  The intervention procedures.**

| groups | intervention | | | | |
|---|---|---|---|---|---|
| | Warming-up | Primary intervention | | | Warming-down |
| **aerobic exercise group** | 5 min (e.g. pedal the cycle ergometer) | cycle ergometer; moderate intensity: 60%-70% of HRmax (HRmax = 220-ages); 20 min | | | 5 min (e.g. pedal the cycle ergometer) |
| **cognitive training group** | 5 min (e.g. sitting quietly) | Naming task(5 min) | calculate task*5 min | working memory task*5 min | color-word stroop task *5 min | 5 min (e.g. sitting quietly) |
| **dual-task group** | 5 min (e.g. pedal the cycle ergometer) | Naming task*5 min | calculate task*5 min | working memory task*5 min | color-word stroop task *5 min | 5 min (e.g. pedal the cycle ergometer) |
| | | cycle ergometer; moderate intensity: 60%-70% of HRmax (HRmax = 220-ages); 20 min | | | |

The calculation task was designed to help improve global EF. A simple 2-number addition or subtraction problem comprising 2-digit numbers was presented on the computer screen for 5 s (e.g., 28 + 14, 39–15), and participants had to provide their answer before the problem disappeared. There were 60 trials in total, and the task lasted 5 min.

The naming task was designed to help develop the ability to shift attention. Participants were asked to speak the names of ordinary items that were displayed at the center of a computer screen for 5 s (e.g., apple, sofa) before the images disappeared. The task comprised 60 trials, each with a different photograph. The task lasted 5 min.

The Stroop color-word task is known to help improve inhibitory control [28]. The name of a color was displayed on the computer screen for 5 s. Sometimes the color of the word's font matched the word itself and sometimes it did not. Participants had to say the color of the word's font (but not the word itself) before it disappeared. There were 60 trials.

The working memory task was used to improve updating ability. A series of numbers (e.g.,14562, 79451) was presented on the computer screen for 2 s, and then faded to the same color as the background for 3 seconds. The participants repeated the series of numbers in threes, and the tester judged whether the answer was correct. The task lasted 5 min.

**2.2.3 Dual-task intervention.** Dual-task intervention combined aerobic exercise and cognitive intervention. The participants were required to pedal the cycle ergometer at a moderate intensity while doing the cognitive tasks.

## 2.3 EF measurements

All EF measures were collected in a light and sound-attenuated room with participants seated in a comfortable chair. Before the intervention, global cognitive state was assessed using the MoCA. Three tests were used to measure the three aspects of EF before intervention, immediately after intervention, and 30 min after intervention.

**2.3.1 Inhibition control.** A Go/No-go task was used to measure inhibition control [29]. A fixation cross was displayed on the computer screen for 300 ms, followed by a visual stimulus at the same location for 700 milliseconds. For Go trials, the word "go" appeared on the screen in green, and the participants had to click the left mouse button with their right index finger. For No-go trials, the word "no go" appeared on the screen in red, and participants had to inhibit their response. The paradigm consisted of 80 Go trials and 40 No-go trials (120 trials total). No-go trials were presented unpredictably by varying the number of intermitting Go trials, with two No-go trials never being presented in succession. The success rate and response time were recorded.

**2.3.2 Information updating.** The ability to update information was evaluated using a 2-back task [30]. A fixation cross was first presented at the center of the screen for 500 ms. Then one of the letters 'B', 'D', 'L', 'Y', or 'P' appeared at random for 500 ms. After the second letter, participants pressed the 'F' key on the keyboard if the current letter was a 2-back match (i.e., the same as the letter presented before the previous letter) and the 'J' key if it was not. For example, in the letter sequence DBYPY, the participant should respond with 'F' following presentation of the underlined letters. The ITI was 1500 ms. There were 30 practice trials before the formal test, which comprised 100 trials. There were three times as many matches as nonmatch. We recorded both the performance and response time.

## 2.4 Rule shifting

The more-odd-shifting task was used to measure shifting ability [31]. The task had three blocks (A, B, C). In each block, a number (1–9, without 5) was displayed at the center of the screen. The A block contained only 'more' trials; the number was always green, and participants were

required to indicate whether it was larger or smaller than 5 by pressing corresponding buttons on the keyboard. The B block contained only 'odd' trials; the number was always red, and participants were required to indicate whether it was odd or even by pressing corresponding buttons on the keyboard. The C Block required shifting; the more-trials and odd-trials were mixed and presented randomly. The C Block had 32 trials, while blocks A and B had 16 trials each. Each block was presented twice in the "ABCCBA" order. A fixation cross was presented during the intertrial intervals (ITIs). In blocks A and B, the numbers were presented for 1100 ms, and the ITI was 900 ms. In block C, the numbers were presented for 1200 ms, and the ITI was 1000 ms. Performance and reaction times were recorded. The 'shifting cost' was calculated by subtracting the average response time on C Blocks from that for A and B blocks.

## 2.5 Statistical analysis

For normally distributed data, continuous variables are reported as mean ± standard deviation. Tests of normality followed by analysis of variance (ANOVA) were used to examine all data. The effects of Time (before, immediately after, 30 min after) and Group (exercise, cognitive training, dual-task) on EF were assessed using as 2-factor ANOVA. Greenhouse-Geisser adjustment was employed when the data did not pass Mauchly's sphericity test. Statistical significance was set to $p < 0.05$.

# 3. Results

## 3.1 Participant characteristics

Fifty-one young men were recruited, and 46 ultimately agreed to participate in the study. They were assigned to the aerobic exercise (n = 15), cognitive training (n = 15), and dual-task (n = 16) groups. **Table 2** (Table 2 baseline characteristics) summarizes the demographic characteristics of the participants. A one-way ANOVA was used to compare demographic data and pre-exercise characteristics among groups to guarantee sample homogeneity. We found no significant differences in age ($p = 0.39$) or global cognitive performance ($p = 0.23$).

## 3.2 Inhibition control

**Table 3** (Table 3 results of EF measurement across intervention groups and time points) summarizes the changes in cognitive function from baseline to follow-up at immediately after intervention and 30 minutes after intervention in three groups.

Inhibition control was assessed using a Go/No-go task. The two-way ANOVA for performance on the Go/No-go task revealed a main effect of Time ($F_{(2,86)} = 3.467$, $p = 0.036$), but no significant Time ×Group interaction ($F_{(4,86)} = 2.377$; $p = 0.058$). Specifically, 30 min after

**Table 2. Baseline characteristics.**

| group | participant (n) | age (year) | MoCA |
|---|---|---|---|
| motor-cognitive dual-task group | 16 | 21.06±3.45 | 26.63±1.41 |
| aerobic training group | 15 | 19.73±2.49 | 27.2±2.08 |
| cognitive training group | 15 | 21.13±3.31 | 26±2.14 |
| total | 46 | 20.65±3.12 | 26.61±1.91 |
| F | | 0.965 | 1.507 |
| p | | 0.39 | 0.23 |

*p<0.05, **p≤0.01; MoCA: Montreal cognitive assessment.

**Table 3. Results of executive function measurement across intervention groups at three time points.**

| | motor-cognitive dual-task group | | | aerobic training group | | | cognitive function group | | |
|---|---|---|---|---|---|---|---|---|---|
| | before intervention | immediately after intervention | 30-min after intervention | before intervention | immediately after intervention | 30-min after intervention | before intervention | immediately after intervention | 30-min after intervention |
| n | 16 | 16 | 16 | 15 | 15 | 15 | 15 | 15 | 15 |
| correct rate of Go/Nogo task (%) | 98.19±2.25 | 98.07±2.29 | 97.03±2.15 | 98.07±2.29 | 97.11±1.96 | 97.83±2.04 | 97.03±2.15 | 97.27±2.36 | 96.12±3.32 |
| reaction time of Go/Nogo tas (ms) | 219.22±31.42 | 227.29±21.56 | 221.03±18.88 | 224.99±19.29 | 216.01±14.3 | 217.45±26.02 | 226.11±24.16 | 236.79±19.55 | 230.82±24.89 |
| correct rate of 2-back task (%) | 32.38±9.80 | 39.50±10.60 | 41.19±18.32 | 32.38±9.80 | 39.50±10.60 | 41.19±18.32 | 32.38±9.8 | 39.50±10.60 | 41.19±18.32 |
| reaction time of 2-back tas (ms) | 445.87 ±153.46 | 420.55±105.15 | 367.32±108.51 | 361.67 ±106.01 | 339.43±85.35 | 338.79±85.44 | 377.39 ±131.84 | 355.32±116.63 | 379.64±101.75 |
| correct rate of shifting task (%) | 50.78±17.00 | 63.28±18.82 | 64.55±15.01 | 45.83±13.60 | 60.63±14.49 | 67.19±11.41 | 43.49±13.38 | 53.54±16.04 | 57.29±14.82 |
| reaction time of shifting tas (ms) | 711.15±37.52 | 684.10±54.24 | 655.62±141.67 | 656.19 ±104.54 | 646.22±100.87 | 652.82±114.12 | 705.13±72.98 | 729.98±75.98 | 698.452±65.49 |
| shifting cast (ms) | 155.34±64.04 | 132.08±53.41 | 124.3±107.38 | 79.43±104.83 | 68.04±147.28 | 104.67±118.37 | 116.12±94.30 | 135.21±59.16 | 90.95±105.47 |

Abbreviations: Values are presented as mean ± SD.

intervention, the performance on the Go/No-go task in the cognitive training group was lower 1.7% than it was before intervention ($p = 0.016$, 95% confidence interval [CI]: 0.003 to 0.032).

The two-way ANOVA for Go/No-go response time did not find any differences among the three groups at any of the three time points ($F_{(1.632,70.19)} = 0.772$; $p = 0.442$). The Time × Group interaction was not significant ($F_{(3.265,70.196)} = 1.867$; $p = 0.138$).

### 3.3 Information updating

The 2-back task was used assess the ability to update working memory. On task accuracy we found a significant main effect of Time ($F_{(2,86)} = 7.05$, $p = 0.001$), it increases over time. We did not observe a significant Time × Group interaction ($F_{(4,86)} = 0.413$, $p = 0.799$). In the dual-task group, performance after 30 min of intervention was greater 8.8% than it was before intervention ($p = 0.038$, 95% CI: 0.4% to 17.3%) and reaction time was lower 78.55ms than before intervention ($p = 0.023$, 95% CI: −148.43 to −8.663).

### 3.4 Rule shifting

The ability to shift between rules was assessed using the more-odd-shifting task. We found a significant main effect of Time ($F_{(2,86)} = 30.698$, $p = 0.01$) on task accuracy, it increases over time, but no significant Time × Group interaction ($F_{(4,86)} = 0.739$, $p = 0.558$). Shifting ability was improved immediately ($p = 0.015$, mean difference = −12.5%, 95% CI: −23% to 1%) and 30 min after ($p = 0.001$, mean difference = −13.8%, 95% CI: −22.3% to −5.2%) the dual-task

intervention. Similarly, it was improved immediately after ($p = 0.005$, mean difference = −14.8%, 95% CI: −25.6% to −3.9%) and 30 min after ($p < 0.001$, mean difference = 21.3%, 95% CI: −30.1% to −12.5%) aerobic exercise. For the cognitive training group, we found that the ability to shift was improved 30 min after intervention ($p = 0.001$, mean difference = −13.8%, 95% CI: −22.6% to 4.9%).

## 4. Discussion

We examined the effects of aerobic exercise, cognitive training, and dual-task intervention on EF in young men. We found that 30-min of intervention improved some aspects of EF, regardless of the type of intervention. Specifically, both 30-min of single-task aerobic exercise intervention and 30-min of dual-task intervention improved attention shifting and working memory updating abilities. We did not find any significance different in inhibitory control. Most importantly, we did not find that dual-task interventions improved EF any more than single-task, aerobic or cognitive, interventions.

### The effect of acute aerobic training on EF

Our results showed a main effect of Time on accuracy for the shifting task. This is consistent with previous research showing that only 30 min of acute moderate-intensity aerobic exercise improved shifting ability in all age groups [32–34]. In addition, like Kamijo et al., our results also indicated a significant improvement in the ability to update working memory following acute aerobic exercise. However, unlike Ji et al. [35], we did not observe improved inhibition control. These mixed results could be related to the intensity of the acute aerobic exercise. Similar to the current findings, Su et al. [36] found no change in inhibitory control or the magnitude of electroencephalographic P3 event-related potentials after acute high-intensity exercise intervention [37].

In our study, we only assessed EF with cognitive tests. However, other studies have used neuroimaging techniques to examine the effects of acute aerobic exercise on brain activity. Functional magnetic resonance imaging (fMRI) and functional near-infrared spectroscopy (fNIRS) evidence has suggested that the prefrontal cortex, which is important for executive function, may be especially sensitive to exercise-induced neurophysiological changes, and that increases in central arousal [38]. Furthermore, exercise-induced upregulation of neurotrophins and neurotrophin-related gene vb expression is thought to be important for synaptic plasticity. Acute aerobic exercise-induced increases in brain-derived neurotrophic factor (BDNF) might explain the improvement in EF after exercise [39]. Furthermore, elevations in insulin-like growth factor 1 (IGF-1) as a result of acute aerobic exercise could work in tandem with BDNF to induce neural plasticity in the brain and improve cognitive performance [40].

Thus, we believe that acute aerobic exercise can improve EF in young men, but it is worth noting that changes in BDNF following acute aerobic exercise are transient [41], and Bezerra et al. [42] also claimed that the gain in EF disappears 60 min after exercise. Therefore, in order to gain better EF, long-term intervention is necessary.

### The effect of acute cognitive training on EF

Although we did not find significant differences in EF after a bout of cognitive training, we observed a trend for information updating and shifting to be improved. Cognitive training improves EF by activating the prefrontal cortex, which is important for EF [43]. Unlike aerobic training or the dual-task, we did not observe significant differences in EF after 30-min cognitive training. The differ results between exercise and cognitive training could be related to BDNF levels. A bout of cognitive training has been shown to have no positive effect on BDNF,

while a bout of aerobic exercise can boost peripheral BDNF concentration [44]. As a result, short-term cognitive training is not recommended because of its limited impact on EF. However, more than one week of cognitive training can increase gray matter volume, and promote task automation [45]. Indeed, long-term cognitive function training (8 weeks) has a positive impact on EF, which improves the quality of life, because over time, cognition can strengthen white matter microstructure [46]. In our study, we found that cognitive training actually led to poorer inhibition control, possibly due to cognitive fatigue.

### The effect of acute dual-task intervention on EF

The dual-task intervention led to improved EF in terms of shifting and working memory abilities, and the effect lasted for up to 30 minutes after the intervention. Similarly, Pellegrini-Laplagne et al. [47] observed that in older people, the effect on flexibility cost was larger for dual-task training than for single-task training. Few studies have explored how acute dual-task intervention improves the ability to update working memory, but long-term dual-task training (12 weeks) has been shown to improve working memory performance in older people [48].

However, as with acute aerobic exercise, we did not find any changes in inhibition control. Indeed, results regarding inhibition control are mixed in the literature. While one study found it beneficial [49], another shows no evidence that inhibitory control is engaged by dual-task intervention [50]. Similarly, neither Pellegrini-Laplagne et al. [47] or Bilgin and Iyigun [51] found a significant improvement in inhibition control after multitask intervention.

According to a recent study, EF deeply regulates motor-cognitive dual-tasks [52] because it is involved in the processes controlling stimulus-category priming and responding. Thus, dual-task training significantly improved shifting task performance. Koch et al. [53] emphasized the fundamental similarity of the underlying cognitive mechanisms between the shifting task and the dual-task. Moreover, dual-task practice contributed to better task coordination skill, which led to improved shifting ability [54]. When participants were trained with a motor-cognitive dual-task, better working memory also helped to improve performance by promoting the updating of cognitive task rules. Overall, the mechanisms through which dual-tasks might improve EF in young men are unclear.

Although our result did not show the difference effect of EF between the single aerobic exercise and the motor-cognitive dual task, there were some variation according to neuroimaging evidence (fMRI, fNIRS), like dual-task elicited larger region of activity in the frontal and parietal lobes than did single-tasks [55], and broader activation in a predominately right-sided fronto-parietal network and the cerebellum than single task [56].

On the other hand, compared with short-term intervention, long-term intervention made difference between single task and dual-task. 12-week dual-task training improved more cognitive ability domains in older people than did single-task training [57]. In addition, long-term intervention motor-cognitive dual-tasks not only improve cognitive function, but can also enhance movement function. After three months of motor-cognitive dual-task training, older people had better balance and faster walking speeds [58]. This improvement might have been due to the release of cognitive resources used to monitor posture when executing the cognitive task during the motor-cognitive dual-task. Researchers thus believe that dual-tasks can promote motion automation [59].

### Limitation

Limitations of the current study include its small sample size, which reduces the power to detect significant effects. Second, we only measured behavior. It will be important in future studies to investigate the dynamic changes in brain activity following motor-cognitive dual-

task training. Although, short-term intervention is an effective way to explore the effect of dual-tasks on the EF of an individual, long-term intervention might be a better way to achieve continuous benefits. It is essential to conduct a large trial to examine the effect of motor-cognitive dual-task training on EF in a wide range of people.

## 5. Conclusion

The major conclusion that can be drawn from this study is that acute aerobic exercise, cognitive training, and dual-task intervention for 30 min can improve the EF performance for young men. Acute aerobic exercise and the dual-task promoted working memory and shifting ability. Moreover, the improvement after dual-task intervention lasted for at least 30 min.

## Acknowledgments

The author is grateful to all participants.

## Author Contributions

**Conceptualization:** Hua Liu.

**Data curation:** Tong Zhang.

**Formal analysis:** Weibin Zhang.

**Funding acquisition:** Weibin Zhang.

**Investigation:** Weibin Zhang.

**Methodology:** Weibin Zhang.

**Project administration:** Weibin Zhang.

**Resources:** Tong Zhang.

**Software:** Weibin Zhang.

**Supervision:** Hua Liu.

**Visualization:** Tong Zhang.

**Writing – original draft:** Weibin Zhang.

**Writing – review & editing:** Hua Liu.

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
