## [Decision Letter · Decision Letter 0]

12 Apr 2023

PONE-D-23-01682Immediate and short-term effects of single-task and motor-cognitive dual-task on executive functionPLOS ONE

Dear Dr. Liu,

Thank you for submitting your manuscript to PLOS ONE. After careful consideration, we feel that it has merit but does not fully meet PLOS ONE’s publication criteria as it currently stands. Therefore, we invite you to submit a revised version of the manuscript that addresses the points raised during the review process (REVIEWER 2).

We look forward to receiving your revised manuscript.

Kind regards,

Luis Carrasco

Academic Editor

PLOS ONE

Journal Requirements:

"This work was supported by North Sichuan Medical College (CBY21-QB14)."

We note that one or more of the authors is affiliated with the funding organization, indicating the funder may have had some role in the design, data collection, analysis or preparation of your manuscript for publication; in other words, the funder played an indirect role through the participation of the co-authors. If the funding organization did not play a role in the study design, data collection and analysis, decision to publish, or preparation of the manuscript and only provided financial support in the form of authors' salaries and/or research materials, please do the following:

(1) Review your statements relating to the author contributions, and ensure you have specifically and accurately indicated the role(s) that these authors had in your study. These amendments should be made in the online form.

(2) Confirm in your cover letter that you agree with the following statement, and we will change the online submission form on your behalf: 

"This work was supported by North Sichuan Medical College, China (CBY21-QB14)."

"This work was supported by North Sichuan Medical College (CBY21-QB14)."

"the author/s claimed no conflict of interest in the present study."

7. Please ensure that you include a title page within your main document. You should list all authors and all affiliations as per our author instructions and clearly indicate the corresponding author.

8. Please amend either the title on the online submission form (via Edit Submission) or the title in the manuscript so that they are identical.

9. Your ethics statement should only appear in the Methods section of your manuscript. If your ethics statement is written in any section besides the Methods, please move it to the Methods section and delete it from any other section. Please ensure that your ethics statement is included in your manuscript, as the ethics statement entered into the online submission form will not be published alongside your manuscript. 

10. Please include a separate caption for each figure in your manuscript.

11. Please ensure that you refer to Figures 1 and 2 in your text as, if accepted, production will need this reference to link the reader to the figure.

**Additional Editor Comments:**

Reviewers have now commented on your paper. You will see that Reviewer 2 is advising that you revise your manuscript (major comments). If you are prepared to undertake the work required, I would be pleased to reconsider my decision.

If you decide to revise the work, please submit a list of changes or a rebuttal against each point which is being raised when you submit the revised manuscript.

Reviewers' comments:

Reviewer's Responses to Questions

**Comments to the Author**

1. Is the manuscript technically sound, and do the data support the conclusions?

Reviewer #1: Yes

Reviewer #2: No

2. Has the statistical analysis been performed appropriately and rigorously? 

Reviewer #1: Yes

Reviewer #2: No

3. Have the authors made all data underlying the findings in their manuscript fully available?

Reviewer #1: Yes

Reviewer #2: No

4. Is the manuscript presented in an intelligible fashion and written in standard English?

Reviewer #1: Yes

Reviewer #2: No

5. Review Comments to the Author

Reviewer #1: Dear Author,

The manuscript entitled "Immediate and short-term effects of single-task and motor-cognitive dual-task on executive function" presents a topic that may interest readers of the journal. 46 male as participants were asked to complete the interventions (aerobic exercise, cognitive training, or dual-task. The results showed that acute aerobic exercise or cognitive training and dual-task interventions play an important role in decision-making processes. I consider an interesting topic, I would like to accept this manuscript.

Reviewer #2: After reviewing your work, my decision is that it requires major corrections.

In this way, I ask the author to answer each of the following questions:

1-The justification of the work is poor. It must be improved.

2- Material and method.

2.1. The calculation of the sample size is not reported.

2.2. Exclusion criteria are not included on: diet, sleep, level of physical exercise practice, academic level, bilingualism, socioeconomic level, etc. that may interfere with the results obtained.

2.2. How the exercise intensity of 60-70% is justified.

23. It is necessary to specify the model of the cycle ergometer, the power that it develops,...

2.4. Why do they always perform the same sequence of cognitive tasks?

2.5. Have you considered that the same sequence in the cognitive tasks can have an influence on the results?

2.6. It is very likely that there is an effect on the outcome of cognitive tasks due to progressive error. How was this effect controlled for?

2.7. The effect size calculation should be included as well as data on the statistical power achieved and for each variable.

3. The objective of the study is not well defined.

4. Minor corrections.

4.1. Improve the writing of the document.

4.2. Correct misspellings (for example, Stoop Test appears instead of Stroop Test) and others.

4.3. Correct typos (graphical).

6. PLOS authors have the option to publish the peer review history of their article (what does this mean?). If published, this will include your full peer review and any attached files.

Reviewer #1: **Yes: **Ebrahim Norouzi

Reviewer #2: **Yes: **Inmaculada Concepción Martínez Díaz

---

## [Author Response · Author response to Decision Letter 0]

19 Jul 2023

Dear reviewers:

Thanks very much for reviewing our manuscript timely and carefully. You have given us lots of good suggestions. We’ve revised our paper according to your comments, and the revised part was marked in red. Followed are our replies to your comments.

Reviewer#1:

Thank you again for your positive comments and valuable suggestions to improve the quality of our manuscript.

Reviewer#2:

Comment 1.the justification of the work is poor.it must be improved.

Answer: we agree with the reviewer’s assessment. Accordingly, throughout the introduction, we have changed the structure of the introduction, and also added more references to explain our research objective. We found there are rare studies to explore the immediate and short-term effect on executive function, and this knowledge will allow us to build a framework for future research into high-efficiency approaches for increasing EF in a variety of people.

Comment 2. material and method 

Comment 2.1 the calculation of sample size is not reported.

Answer: Thank you for this suggestion. We used the G*power to calculate the sample size, the sample size is 45. This work was done in a small sample size due to difficulty in recruiting subjects during the COVIN-19 period. our sample size only included 46 participants. we will added the calculation of sample size this part in the first paragraph in 2.1 Participants.

Comment 2.2 exclusion criteria are not included on: diet, sleeping, level of physical exercise practice, academic level, bilingualism, socioeconomic level, etc. that may interfere with the results obtained.

Answer: Thank you for this suggestion. We added more inclusion criteria in 2.1 participants to describe the participants in this study, and gave more detail in sleeping, the level of physical activity, the ability of education which can influent the cognitive performs. 

Changes: page 2, 2.methods, 2.1 participants 

Comment 2.2 how the exercise intensity of 60~70% is justified.

Answer: Thank you for pointing this out. previously, many researches has reported the acute aerobic exercise with moderate intensity improve executive function significantly.

[1].Kaminsky, L. A. (2014). ACSM's Guidelines for Exercise Testing and Prescription

[2]Tsukamoto, H., Takenaka, S., Suga, T., Tanaka, D., Takeuchi, T., Hamaoka, T., . . . Hashimoto, T. (2017). Impact of Exercise Intensity and Duration on Postexercise Executive Function. Med Sci Sports Exerc, 49(4), 774-784. https://doi.org/10.1249/mss.0000000000001155

[3]Effects of Acute Exercise Duration on the Inhibition Aspect of Executive Function in Late Middle-Aged Adults.We added explanation about the intensity and duration of the aerobic exercise protocol and some references in 2 methods 2.2.1 Aerobic exercise intervention.

Comment 2.3 it is necessary to specify the model of the cycle ergometer, the power that it develops,…. 

Answer: Thank you for pointing this out. We added the model of the bicycle ergometer, and we also added more information about the heart rate monitor.

Changes: all this information has added in 2.2.1 aerobic exercise intervention.

Comment 2.4 why does they always perform the same sequence of cognitive task?

Answer: Thank you for pointing this out. According to previous studies, four cognitive training tasks were designed in this study. Since the block of each task only appeared once, the subjects would complete these fore cognitive training tasks according to the difficulty of the task. (refrence: Rezola-Pardo, C., Arrieta, H., Gil, S. M., Yanguas, J. J., Iturburu, M., Irazusta, J., Rodriguez-Larrad, A. (2019). A randomized controlled trial protocol to test the efficacy of a dual-task multicomponent exercise program in the attenuation of frailty in long-term nursing home residents: Aging-ON(DUAL-TASK) study. BMC Geriatr, 19(1), 6. https://doi.org/10.1186/s12877-018-1020-z)

Comment 2.5 have you considered that the same sequence in the cognitive task can have an influence on the results?

Answer: Thank you for pointing this out. I am not very sure you are talking about the sequence of these training cognitive test or these three cognitive tasks for the executive function measure. For the training task, they only appeared once, and did not appear again, so we don’t think it can have influence on the results. But if you are talking about the three tasks for executive function test, we think it may had some influence on the results.

Comment 2.6 it is very likely that there is an effect on the outcome of cognitive tasks due to the progressive error. How was this effect controlled for?

Answer: Thank you for point this out. In this study, we tried to controlled all the progressive errors. At first, all the participants had been told the rules of each task, and they also had the practice trials to make sure they totally understand how to complete these tasks. 

Comment 2.7 the effect size calculation should be included as well as data on the statistical power achieved and for each variable.

Answer: Thank you for this suggestion. We added more detail to describe the data in result. We added the mean difference and 95%IC.

Comment 3 the objective of the study is not well defined.

Answer: we appreciate the reviewer’s feedback. the objective of the study was further elaborated in the background. 

Comment 4 minor corrections.

Comment 4.1 improve the writing of the document.

Answer: Thanks for your comment. We tried our best to improve the manuscript and made some change to the manuscript. This change will not influence the content and framework of the paper. And here we did not list the changes. This manuscript was edited by a native English speaker with a PhD in neuroscience.

Comment 4.2 correct misspelling.

Answer: Thank you for pointing this out. The “stoop” has been corrected on the table 1.

Comment 4.3 correct typos.

Answer: We are sorry for spelling mistakes and grammatical errors caused by our carelessness. In the revised version, we have made significant efforts to remove the mistakes and errors and improved writing. All the errors you picked and recommendations you proposed are greatly helpful for us to polish our manuscript. We appreciate your elaborate efforts in reviewing. Thank you very much!

We do appreciate your work in reviewing this paper. We hope these careful changes can make this paper more scientific and readable, and we really hope you can reconsider acceptance of this paper, because it really stands for our efforts and do contribute to the research of sports science.

Sincerely yours,

Hua Liu.

Capital University of Physical Education and Sports,

Email: liuhua@cupes.edu.cn

---

## [Decision Letter · Decision Letter 1]

3 Aug 2023

Immediate and short-term effects of single-task and motor-cognitive dual-task on executive function

PONE-D-23-01682R1

Dear Dr. Liu,

We’re pleased to inform you that your manuscript has been judged scientifically suitable for publication and will be formally accepted for publication once it meets all outstanding technical requirements.

Kind regards,

Thiago P. Fernandes, PhD

Academic Editor

PLOS ONE

Additional Editor Comments (optional):

Reviewers' comments:

Reviewer's Responses to Questions

**Comments to the Author**

1. If the authors have adequately addressed your comments raised in a previous round of review and you feel that this manuscript is now acceptable for publication, you may indicate that here to bypass the “Comments to the Author” section, enter your conflict of interest statement in the “Confidential to Editor” section, and submit your "Accept" recommendation.

Reviewer #1: All comments have been addressed

Reviewer #2: All comments have been addressed

2. Is the manuscript technically sound, and do the data support the conclusions?

Reviewer #1: Yes

Reviewer #2: Yes

3. Has the statistical analysis been performed appropriately and rigorously? 

Reviewer #1: Yes

Reviewer #2: Yes

4. Have the authors made all data underlying the findings in their manuscript fully available?

Reviewer #1: Yes

Reviewer #2: Yes

5. Is the manuscript presented in an intelligible fashion and written in standard English?

Reviewer #1: Yes

Reviewer #2: Yes

6. Review Comments to the Author

Reviewer #1: Dear Authors,

I read the revised manuscript. The authors have addressed my concerns completely, Thank you, in my opinion, it is now can be publish.

Reviewer #2: I consider that with the changes you have made to your work, it is more in line with the demands of scientific journals

7. PLOS authors have the option to publish the peer review history of their article (what does this mean?). If published, this will include your full peer review and any attached files.

Reviewer #1: **Yes: **Ebrahim Norouzi

Reviewer #2: **Yes: **Inmaculada C Martínez-Díaz

---

## [Editor Report · Acceptance letter]

7 Aug 2023

PONE-D-23-01682R1 

Immediate and short-term effects of single-task and motor-cognitive dual-task on executive function 

Dear Dr. Liu:

I'm pleased to inform you that your manuscript has been deemed suitable for publication in PLOS ONE. Congratulations! Your manuscript is now with our production department. 

Kind regards, 

on behalf of

Dr. Thiago P. Fernandes 

Academic Editor

PLOS ONE